# AAV-CRISPR/Cas9 Gene Editing Preserves Long-Term Vision in the P23H Rat Model of Autosomal Dominant Retinitis Pigmentosa

**DOI:** 10.3390/pharmaceutics14040824

**Published:** 2022-04-09

**Authors:** Saba Shahin, Hui Xu, Bin Lu, Augustus Mercado, Melissa K. Jones, Benjamin Bakondi, Shaomei Wang

**Affiliations:** Board of Governors Regenerative Medicine Institute, Department of Biomedical Sciences, Cedars-Sinai Medical Center, Los Angeles, CA 90048, USA; saba.shahin@cshs.org (S.S.); hui.xu@utah.edu (H.X.); bin.lu@cshs.org (B.L.); atmercado25@gmail.com (A.M.); mkaye23@hotmail.com (M.K.J.)

**Keywords:** autosomal dominant retinitis pigmentosa, Rhodopsin P23H, CRISPR/Cas9, AAV delivery, allele-specific ablation, photoreceptors, rod cells, ER-stress, autophagy, long-term vision preservation

## Abstract

Retinitis pigmentosa (RP) consists of a group of inherited, retinal degenerative disorders and is characterized by progressive loss of rod photoreceptors and eventual degeneration of cones in advanced stages, resulting in vision loss or blindness. Gene therapy has been effective in treating autosomal recessive RP (arRP). However, limited options are available for patients with autosomal dominant RP (adRP). In vivo gene editing may be a therapeutic option to treat adRP. We previously rescued vision in neonatal adRP rats by the selective ablation of the Rhodopsin S334ter transgene following electroporation of a CRISPR/Cas9 vector. However, the translational feasibility and long-term safety and efficacy of ablation therapy is unclear. To this end, we show that AAV delivery of a CRISPR/Cas9 construct disrupted the Rhodopsin P23H transgene in postnatal rats, which rescued long-term vision and retinal morphology.

## 1. Introduction

Retinitis pigmentosa (RP) is the most common hereditary retinal disease, characterized by progressive photoreceptor (PR) degeneration, resulting in vision loss and blindness [1,2]. More than 100 genetic loci in over 60 genes have been linked to RP by autosomal dominant, recessive, or X-linked inheritance patterns [3].

Gene augmentation or replacement therapy represents a rational drug design for treating autosomal recessive RP (arRP) and has shown clinical benefit; *RPE65* gene replacement (Luxturna^TM^) received United States FDA approval for Leber congenital amaurosis (LCA) 2 [3,4]. Notably, this was the first approved gene therapy for a hereditary disease, the first using adeno-associated virus (AAV) for RP. However, unlike recessive RP, which is characterized by protein deficiency, adRP requires protein reduction to alleviate toxicity. As a result, classic gene replacement therapy is ineffective for adRP, as they are caused by dominant-negative or gain-of-function mutations. Transcriptional suppression of dominant RP alleles has been shown to rescue vision in adRP animals using ribozymes, RNAi, and zinc-finger nucleases [5,6,7,8]. However, transcriptional silencing is typically transient and allele independent. Genomic disruption as a means to permanently and selectively silence dominant alleles may be a therapeutic option for adRP [9,10,11,12]. Endogenous DNA repair through non-homologous end joining (NHEJ) after double strand breaks generates frameshift mutations, which lead to functional ablation. Targeting endonucleases to adRP mutations using Clustered Regularly Interspaced Short Palindromic Repeat (CRISPR)/Cas9 is being investigated in preclinical studies [10,11,13]. Precedence for its clinical use was shown by FDA approval of CRISPR-based therapies in a handful of clinical studies, including a trial for recessive LCA10 [14,15].

Gene therapies aim to prevent or slow photoreceptor loss, the effectiveness of which depends on early intervention. These therapies will likely impact infants, children, and adolescents in efforts to preserve the maximum amount of existing vision. To this end, we previously treated neonatal adRP rats (S334ter) to demonstrate proof of concept for CRISPR ablation therapy in vivo. Similar to LCA2 patients who frequently show early onset presentation (age 3–7), S334ter line-3 rats represent severe and early onset adRP with rapid photoreceptor degeneration culminating by postnatal day (P) P21. The window of opportunity for treatment is limited from embryonic day (E) E16 to P2. Because the rate of retinal degeneration in adRP patients is heterogeneous, we sought to determine whether ablation therapy can preserve vision in animals that represent older patients. For this, we used a slower degenerating adRP model bearing the most common adRP mutation (rhodopsin P23H) for clinical relevance. P23H (*RHO*^P23H^*)* is the most frequent mutation in the rod cell-specific gene rhodopsin (*RHO*) in adRP patients and is a potent target for mutation-specific ablation strategies using CRISPR [10,13]. The P23H mutation results in rhodopsin misfolding (class II mutation) and mistrafficking (class I mutation), with excessive accumulation in the endoplasmic reticulum (ER) [14,15,16]. P23H line-3 rats undergo photoreceptor degeneration from P15. This allowed us to observe the extent of vision rescue over timeframes commensurate to decades in patients and permitted treatment at multiple stages of disease. In the current study, we showed that subretinal delivery of AAV-CRISPR/Cas9 is safe and that it rescued photoreceptors and vision for 15 months in Rhodopsin P23H-3 rats.

## 2. Materials and Methods

### 2.1. AAV-SaCas9/gRNA Vector Design

Cas9 from *Staphylococcus aureus* (SaCas9) was used to accommodate the size limitation of AAV2/8 particles. gRNA was cloned into pAAV2 vector (Vector Biosciences, Cambridge, MA, USA) via BsaI restriction enzyme sites upstream of the scaffold gRNA sequence, and the mCherry reporter (Addgene, Cambridge, MA, USA) was cloned downstream of the Cas9 transcript regulated by the cytomegalovirus (CMV) promoter. The in vitro assay was used to determine cleavage efficiency per our previous publication [9].

### 2.2. Animals and Injection Procedures

P23H rats were obtained from RRRC; Long Evans (LE) rats and Sprague–Dawley (SD) rats were purchased from Charles River, housed and maintained at the Cedars-Sinai Medical Center Department of Comparative Medicine vivarium. All animal procedures were performed following the Cedars–Sinai Medical Center’s Institutional Animal Care and Use Committee and the ARVO Statement for the Use of Animals in Ophthalmic and Vision Research. P23H/P23H, P23H/LE (Long-Evans rat background), and P23H/SD (Sprague–Dawley rat background) rats, both sexes received unilateral subretinal injection of AAV-SaCas9/gRNA (2 uL, 3.9 × 10^13^ GC/mL) at the age of postnatal day P15 and P28–P29. The controls include AAV8.CMV-mCherry or AAV-CMV-GFP treated and untreated. The reason for using P23H/LE rats was to test visual acuity by optokinetic response on pigmented rats, whereas P23H/SD rates served to model accelerated degeneration.

### 2.3. Visual Function Assessment

Spatial visual acuity and visual functions were evaluated by the optokinetic response (OKR) using OptoMotry testing apparatus (Cerebral Mechanics, Lethbridge, AB, Canada) and electroretinography (ERG) according to our published protocols, respectively [16,17]. Briefly, OKR assessment comprises four computer monitors arranged in a square to project a 3D virtual space of a rotating cylinder lined with vertical sine-wave grating with an unrestrained animal on a center platform tracking the projected image of rotating grating with reflexive head movements. The spatial frequency of the grating (cycles per degree) was centered on the rats’ viewing position, and maximal acuity was ascertained by increasing the grating frequency at psychophysics staircase progression until the tracking response was lost. OKR offers non-invasive screening to detect visual acuity. Additionally, ERG provides a gross measure of retinal activity and indicates the relative function of the retina. Animals were dark-adapted for at least 14–16 h before recordings. The white stimuli were given at an intensity of 25 cd/m^2^ by a computer-controlled system using Espion system (Diagnosys LLC, Lowell, MA, USA). A total of 20–30 sweeps for each stimulus per animal were recorded, and the average responses were used as the response amplitude.

### 2.4. Retinal Cell Isolation, FACS-Based Sorting and Targeted Deep Sequencing

Animals were sacrificed at P36 and P162; eyes were surgically removed and enucleated. Retinas were dissected out and subjected to single-cell dissociation by incubating for 20 min at 37 °C in enzymatic digestion solution consisting of Ca^2+/^Mg^2+^-free phosphate-buffered saline, 20 U/mL papain, and 0.5 mmol/L l-cysteine (Worthington Biochemical Corp., Lakewood, NJ, USA). To confirm the allele-specific disruption of *m-Rho*^P23H^ by SaCas9/gRNA in vivo, PRs expressing SaCas9 were sorted from retinal cell suspensions of AAV-treated P36 P23H/SD rats by SaCas9-HA fusion protein expression using HA.11-FITC. Genomic DNA was isolated from these sorted PRs and processed for PCR-amplification using primers flanking the predicted Cas9 cut site regions and Sanger sequencing. Further, mCherry positive and negative retinal cells were sorted (FACSAria III, BD Biosciences, Franklin Lakes, NJ, USA) from retinal cell suspensions from P162 homozygous (P23H/P23H) and heterozygous (P23H/SD and P23H/LE) P23H rats into 1.7 mL DNase-free tubes (Eppendorf, Hamburg, Germany) containing 4 °C phosphate-buffered saline and subjected to genomic DNA isolation (Purelink Genomic DNA Mini Kit, Life Technologies). The on- and off-target effects of used sgRNA was evaluated by NGS (GENEWIZ, South Plainfield, NJ, USA). Briefly, after performing PCR amplification using primers that flanked the on- and off-target loci (Appendix A), PCR products were used for an NGS assay. NGS data were analyzed using CRISPResso2 software [18]. Gene editing efficiency was determined by calculating the percentage of paired reads for modified/edited and unmodified/unedited alleles (Gene editing efficiency = (modified reads/total target reads) × 100). Potential off-target sites were determined using Cas-OFFinder.

### 2.5. RNA Extraction and qPCR Analyses

Total RNA from P162 and P450 P23H rat retinas was isolated using QIAGEN RNeasy Mini kit (Qiagen Hilden, Germany). cDNA was synthesized in a 20 μL reaction using 500 ng total RNA and ProtoScript^®^ II First Strand cDNA Synthesis Kit (New England Biolabs Inc., MA, USA). Primers were synthesized by Integrated DNA Technologies (San Diego, CA, USA) (Appendix A). Quantitative real-time PCR was performed with PowerUp™ SYBR™ Green Master Mix (Applied Biosystems). The transcript levels of target genes were assessed using Bio-Rad CFX384 Real-Time PCR Detection System (Bio-Rad, Hercules, CA, USA). Cycling parameters were: 95 °C for 2 min, 95 °C for 15 s, 60 °C for 15 s, 72 °C for 30 s, with 40 cycles of amplification. The relative mRNA levels were normalized to the housekeeping gene, GADPH. Differential expression was determined using the deltaCt method.

### 2.6. Histologic Assessment, Immunofluorescent Staining, and Confocal Microscopy

Histological assessment and immunofluorescence were performed as described in our previous publications [9,16,19]. Briefly, the animals were euthanized, eyes were enucleated, and a small cut was made at the center of the cornea. Eyeballs were then fixed in 4% paraformaldehyde in PBS for 1 h at room temperature, transferred to 10%, 20%, and 30% sucrose (1 h for each concentration at room temperature) and kept in 30% sucrose overnight at 4 °C. The next day, eyeballs were embedded in optimal cutting temperature (OCT) compound (Sakura Finetek, Torrance, CA, USA) and stored at −80 °C. Frozen transverse 10 μm thick retinal sections were collected in five series with 4 sections per slide. One set of slides was stained with cresyl violet (0.4% cresyl violet acetate, Sigma Aldrich, St. Lois, MO, USA) to evaluate retinal lamination and photoreceptor preservation. The remaining slides were stained with immunofluorescence. Antibodies used: cone arrestin (rabbit polyclonal, 1:1000, AB15282, Millipore, Billerica, MA, USA), Cathepsin D (goat polyclonal, 1:100, sc-6486, Santa Cruz Biotechnology), glutamine synthetase (clone GS-6, 1:1000, MAB302, Millipore, Billerica, MA, USA), Glial Fibrillary Acidic Protein (G-A-5, mouse monoclonal, 1:1000, G3893, Sigma), LC3B (rabbit polyclonal, 1:200, NB100-2220, Novus Biologicals), protein kinase C-α (rabbit polyclonal, 1:8000, P4334, Sigma), recoverin (rabbit polyclonal, 1:2000, AB5585, Millipore, Billerica, MA, USA), rhodopsin (C-terminal, clone 1D4, 1:100, MAB5356, Millipore, Billerica, MA, USA), Synaptophysin (mouse clone SVP-38, 1:2000, Millipore MAB368), and P62/SQSTM1 (mouse monoclonal, C11, 1:1000, H00008878-M01, Novus Biologicals). Alexa-Fluor-488 or Cy3 secondary antibodies (1:500; Life Technologies) were used to visualize sections along with nuclear counterstain (DAPI, 49,69-diamidino-2-phenylindole; Vector Laboratories, Burlingame, CA, USA). For antibody specificity, controls included omitting the primary or secondary antibodies, which leads to negative staining. Images were captured by confocal microscope (Zeiss LSM710, Oberkochen, Germany) and stored as TIFF files. For quantifying integrated fluorescence intensity (also known as Integrated Optical Density-IOD) for each antigen, images from both AAV-treated and untreated retinal sections were acquired at same magnification using same laser power (exposure), gain and offset.

### 2.7. Image Analysis

Distribution of mCherry signals, PR nuclei counts, synaptic puncta counts and IOD measurements was performed using Image J software (Image J 1.53 bundled with 64-bit Java, NIH, Bethesda, MD). For mCherry distribution in retinal whole mount, the percent area showing mCherry positive signals was measured after normalizing to the background signal and subtracting the background area in retinal whole mount. The number of mCherry positive cells, PRs nuclei and synaptic puncta were counted using ‘threshold’ and ‘analyze particles’ functions. Differences in immunoreactivity of Cone arrestin, Cathepsin D, GS, GFAP, LC3B, PKC-α, Recoverin, Rhodopsin, Synaptophysin and P62/SQSTM1 in both AAV-treated and untreated retina were assessed as integrated optical density (IOD), as mentioned elsewhere [20,21]. In brief, for IOD measurements, region of interest (ROI) for each protein was selected by using free hand tool in ImageJ. Fluorescent intensities of ROIs were measured after normalizing to the background. The fluorescent intensities (IOD values) were averaged (4 sections per retina and 4 retina per group; each section 40 μm apart) to determine signal density. The IOD values were presented as arbitrary units (a.u.). The processing and analysis were kept consistent between images. Statistical analysis was performed on average IOD values (mean ± SEM). Fold change in signal intensity for each antigen was calculated by normalizing the average IOD values of AAV-treated retina to the untreated retina. Based on differences in the level of significance (*p* value < 0.05), the IOD values were classified as intense, moderate, or weak. Image analysis was performed blindly and separately by 2 observers.

### 2.8. Statistical Significance

Student’s *t* tests were performed using two-tailed distribution and two-sample unequal variance (heteroscedastic) to compare optokinetic response cycles/degree values from treated versus untreated eyes of individual animals, as well as between animals of treatment groups. Contralateral noninjected eyes served as untreated controls. Analyses were run using GraphPad Prism. Values are expressed as mean ± SEM. Statistical significance: * *p* ≤ 0.05, ** *p* ≤ 0.01, *** *p* ≤ 0.001, **** *p* ≤ 0.0001.

## 3. Results

### 3.1. Guide RNA Vector Design Strategy

The AAV2-SaCas9/gRNA vector design is shown in Figure 1A. The mCherry was driven by the cytomegalovirus (CMV) promoter, SaCas9 was driven by a mouse rhodopsin proximal promoter (mOP500) [22], and the gRNA was driven by a human U6 promoter. The construct size between ITRs was 4.75 kb, and 6.4 kb including mCherry. The P23H point mutation was located in exon 1 (Figure 1B; red font). We designed three 23-nucleotide targeting-guide RNA (gRNA) constructs complementary to region in exon 1 immediately upstream of PAMs unique to the mouse Rho locus to discriminate mouse and rat alleles during SaCas9 cleavage (Figure 1B). SaCas9/gRNA cleavage efficiency was assessed for each gRNA by T7E1 endonuclease assay and confirmed by Sanger sequencing. Cleavage efficiency of the gRNAs was tested in vitro, in bone marrow-derived mesenchymal stem cells isolated from P23H rats using methods previously reported [9]. The homology between mouse transgene and rat Rho alleles at the selected gRNA binding locus differed by one nucleotide in the PAM sequence. Further, to evaluate AAV toxicity and decide the optimal dose, P23H/SD rats received a single unilateral trans-scleral subretinal injection of 2 µL AAV at different concentrations (Appendix A) at P15. To check the retinal response, we performed ERG on these rats at P90. ERG data at P90 showed significantly reduced visual function in perturbed retina from AAV controls, whereas retina treated with AAV-SaCas9/gRNA not only demonstrated improved visual function but also had no deleterious effect compared to sham-operated eyes (Appendix A).

### 3.2. SaCas9/gRNA Distribution in Retinas of P23H Rats following Subretinal Injection

We examined the distribution of mCherry expression on retinal whole-mount preparation at the early (P42) and late (P162) time points after injection. We observed robust mCherry expression at both time points (Figure 2A). The reason for continued mCherry expression is incorporated in the Discussion section. mCherry expression served as a marker for expression of SaCas9 and sgRNA, and high-power microscopic images showed strong mCherry expression in photoreceptors. In the retinal whole mount, mCherry distribution was 52% at P42 and 49% at P162 with uneven fluorescence intensity in transfected regions (Figure 2A, upper panel). By cross-section analysis, the density of PRs in mCherry-positive retinal regions at P42 was 3387 nuclei/50 mm and 2090 nuclei/50 mm at P162 (Figure 2A, lower panel). The presence and distribution of mCherry expression were documented from 3 to 20 weeks after subretinal delivery. Next, to confirm allele-specific disruption of *m-Rho*^P23H^ by SaCas9/gRNA in vivo, neural retinas were collected 3 weeks after treatment at P15 in P23H/SD rats, processed for enzymatic single-cell dissociation. SaCas9 expressing photoreceptors (PRs) were sorted by FACS based on the expression of SaCas9-HA fusion protein using HA.11-FITC (Figure 2B(i,ii)) and processed for Sanger sequencing, which confirmed *m-Rho*^P23H^ disruption downstream of the gRNA binding site in treated eyes (Figure 2C(i), shaded region), but not in untreated (control) eyes (Figure 2C(ii), shaded region). The presence of overlapping base calls with decreased nucleotide read fidelity (Phred quality scores) (Figure 2C(i)) indicated the generation of genomic variants by indels during DNA repairs following NHEJ. Further, we also performed qPCR of *m-Rho*, *r-Rho* and *r-Gnat1* (a-transducin) genes in retinal samples following *m-Rho*^P23H^ ablation to check the alterations in transcript levels of these genes in the overall retina, as only a part of the retina was affected by AAV treatment. mRNA expression of *r-Rho* increased by 3.54-fold (*p* ≤ 0.01) while 1.61-fold reduction in *m-Rho* (*p* ≤ 0.01) mRNA expression was detected in AAV-treated retina (Figure 2D). The mRNA expression of *r-Gnat1*, a specific marker gene for rod cells that encodes the α subunit of rod transducin, a key signaling molecule in the rod phototransduction cascade [23], was increased by 5.04-fold (*p* ≤ 0.001) in treated retina compared to untreated (Figure 2D), suggesting the restoration of a rod phototransduction signaling pathway following m-Rho^P23H^ ablation.

### 3.3. On- and Off-Target Genome Editing Analysis following In Vivo m-Rho^P23H^ Disruption

For in vivo testing, both homozygous (P23H/P23H, carrying transgene insertions at both alleles) and heterozygous (P23H/SD or P23H/LE, carrying transgene insertion at single allele) P23H rats received a single unilateral trans-scleral subretinal injection of AAV-SaCas9/gRNA. After 20 weeks of AAV treatment, m-Cherry positive and negative cells were isolated from retinal cell suspensions through FACS and processed for Next-Generation Sequencing (NGS) to evaluate the on- and off-target effects. Targeted deep sequencing analysis revealed genome editing efficiency of 5.97% for P23H/P23H (Figure 3A(i)), 12.08% for P23H/SD (Figure 3A(ii)) and 14.8% for P23H/LE (Figure 3A(iii)) mCherry positive cells, while no gene editing was detected in mCherry-negative cells (Appendix A, Appendix A). In AAV-treated eyes, m-*Rho*^P23H^ editing showed insertions and/or deletions (indels) at the predicted cleavage site (Figure 3B,C). Further, we also observed that SaCas9/gRNA-mediated indels lead to frameshift in the targeted coding region of m-*Rho*^P23H^ (Appendix A). Mismatch motif candidate targets were selected using an off-target prediction tool [24] to analyze off-target effects of SaCas9/gRNA, and each potential off-target amplicon was subjected to NGS, and no gene modification was detected (Appendix A). Furthermore, no on- and off-target gene editing was detected in untreated retinas (Appendix A).

### 3.4. Long-Term Vision Rescue following m-Rho^P23H^ Ablation

To assess the long-term efficacy of allele-specific ablation therapy, we performed ERG at multiple time points over 12 months after single subretinal injection of AAV-SaCas9/gRNA. AAV-treated eyes had higher ERG a- and b-wave amplitudes for up to 12 months (Figure 4A). Statistical analysis showed that both a-wave and b-wave amplitudes in the AAV-treated eyes were higher than the contralateral control eyes at P60 (a- and b-wave: *p* ≤ 0.01), P90 (a- and b-wave: *p* ≤ 0.001), P120 (a- and b-wave: *p* ≤ 0.001), P150 (a- and b-wave: *p* ≤ 0.001) and P200 (a-wave: *p* ≤ 0.01; b-wave: *p* ≤ 0.05) for P23H/P23H rats (Figure 4A, upper panel, *n* = 12); P180 (a-wave: *p* ≤ 0.05), P209, P249, P279 (a- and b-wave: *p* ≤ 0.05) and P361 (b-wave: *p* ≤ 0.05) for P23H/SD rats (Figure 4A, middle panel, *n* = 8); and P93, P240 (b-wave: *p* ≤ 0.05) and P361 (a- and b-wave: *p* ≤ 0.05) for P23H/LE rats (Figure 4A, lower panel, *n* = 12). It is noted that the amplitudes of both a-and b-waves have never reached wild-type level (Appendix A). Our previous study has shown that rod photoreceptor function was compromised at the early stage of disease [25]. Gene editing targeted to rods did not rescue rod function to normal levels in the current study. However, progressive retinal degeneration was slowed down significantly. Further, we determined the visual acuity in P23H/LE rats by measuring the optokinetic response (OKR). Visual acuity of P23H/LE rats assessed at 15 months was two-fold higher in AAV-treated eyes compared with untreated contralateral eyes (0.276 versus 0.137 cycles/degree, *p* ≤ 0.01, *n* = 5; Figure 4B). It is noted that preserved visual acuity is about half of the wild-type rat value (0.500–0.600 c/d).

### 3.5. Histologic Evaluation of Retina following m-Rho^P23H^ Ablation

Cresyl violet-stained retinal sections were used to assess retinal lamination and PR preservation. The outer nuclear layer (ONL) is composed of PR nuclei, approximately 97% of which are rods and 3% are cones in rodents [26]. In untreated retina and retinal regions distal to the AAV injection site, the ONL was reduced to one cell layer at 15 months of age (Figure 4C). Areas of the retina close to AAV injection had a significantly higher number of PR nuclei (P23H/P23H: 1–4 layers, PR nuclei = 217 ± 19/100 μm, *p ≤* 0.001, *n* = 11; P23H/SD: 1–4 layers, PR nuclei = 414 ± 24/100 μm, *p ≤* 0.0001, *n* = 11; P23H/LE: 4–5 layers, PR nuclei = 524 ± 50/100 μm, *p ≤* 0.001, *n* = 11) in the ONL (Figure 4C and Appendix A). The ONL preservation was evident over 2600 µm along the rostral–caudal axis (Appendix A). Taken together, these data suggest that CRISPR-Cas9 mediated allele-specific disruption of *m-Rho^P23H^* significantly improved light sensitivity and visual acuity through PR preservation compared with untreated retina.

### 3.6. Long-Term Preservation of Photoreceptors, Rod/Cone-Bipolar Dendritic Arborization, and Retinal Synaptic Connections following m-Rho^P23H^-Specific Ablation

To evaluate the long-term effects of ablation therapy in preserving PRs and inner retinal synaptic connections, we processed retinal sections from P450 rats for immunofluorescence using various retinal cell markers. AAV-treated regions of the retina were identified by mCherry expression and showed rhodopsin immunoreactivity in the outer segments (OS) (fold change: 2.4, *p ≤* 0.0001, *n* = 4; Figure 5A inset: shown by arrow, green: rhodopsin, red: mCherry), suggesting that *m-Rho*^P23H^ ablation corrected *Rho^WT^* trafficking to OS. By contrast, rhodopsin immunoreactivity was restricted to PR cell bodies in untreated retina, suggesting ER retention of rhodopsin due to trafficking failure (Figure 5A, inset: arrow, green: rhodopsin). Immunolabeling with cone arrestin revealed the whole profile of cones with organized segments and pedicles that were well preserved for 15 months in AAV-treated eyes. Untreated retina contained predominantly fragmented, irregular-shaped and degenerative cones, indicative of apoptotic bodies (fold change: 1.4, *p ≤* 0.05, *n* = 4, Figure 5B). Recoverin expression in PRs and ON cone bipolar cells was significantly increased (fold change: 4.1, *p ≤* 0.01, *n* = 4, Figure 5C). PKCa labeled rod bipolar cells and synaptic connections between PRs and retinal ganglion cells and were preserved significantly in AAV-treated retina, compared with control (fold change: 1.8, *p ≤* 0.001, *n* = 4, Figure 5D). Furthermore, PKCa immunostaining revealed pseudo-stratified cell bodies of bipolar cells with dense dendritic arborization and axon terminals in AAV-treated retina.

By contrast, disorganized bipolar cells with reduced density of axon terminals were observed in untreated retina. In addition to the long-term preservation of PRs and bipolar cells, we observed a denser synaptic connection between PRs and inner retinal neurons in the outer plexiform layer (OPL) of AAV-treated retina as revealed by synaptophysin expression (Figure 5E, inset: arrow, green: synaptophysin; red: mCherry). Significantly higher expression of synaptophysin was also observed in the inner plexiform layer (IPL) region of AAV-treated retina (fold change: 1.8, *p ≤* 0.01, *n* = 4, Figure 5E). Moreover, preserved synaptic connections and synaptic density, indicated by significantly higher synaptic density as indicated puncta, were also observed in the OPL of AAV-treated retina (fold change: 3.4, *p ≤* 0.01, *n* = 4, Figure 5E). Collectively, these data demonstrated that PRs and their retinal connection were well preserved for up to 15 months after gene editing.

### 3.7. Suppression of Gliosis and Phagocytic Activity in Müller Glia following m-Rho^P23H^ Ablation

Gliosis in Müller glia is a hallmark of retinal degeneration [27]. The phagocytic activity of Müller glia eliminates degenerative PRs and cell debris after retinal injury [28,29]. Increased phagocytic activity of activated Müller glia was also reported in the retina of animals homozygous for the P23H mutation [29]. Accordingly, we assessed Müller cells activation (gliosis) and phagocytic activity in *m-Rho^P23H^*-bearing rat retinas. Gliosis and phagocytic activity in Müller cells was decreased significantly in AAV-treated retina. Glutamine synthetase (GS) immunostaining in retinal sections of P450 AAV-treated rat eyes revealed well-preserved organotypic morphology of Müller glia with singular thin processes that spanned the ONL and encircled mCherry-positive PRs to support retinal homeostasis (fold change: 1.1, *p ≤* 0.05, *n* = 4, Figure 6A). By contrast, we observed Müller glia with thick and elongated processes that surrounded degenerative PRs (condensed nuclei visualized as intense DAPI) (Figure 6A inset: arrows) in control. Further, significantly decreased GFAP expression in Müller glia end-feet strongly suggested the predominantly quiescent Müller glia in AAV-treated retina. Conversely, we observed strong GFAP expression in the contralateral retina, devoid of mCherry, suggesting robust activation of Müller glia (fold change: 5.2, *p ≤* 0.01, *n* = 4, Figure 6B).

### 3.8. m-Rho^P23H^ -Specific Ablation Rescues PRs and Preserves Long-Term Vision by Suppressing UPR/ERS-Mediated Autophagy

Corrected rhodopsin trafficking preserved PRs and long-term vision by suppressing ERS/UPR and inhibiting autophagy-mediated apoptosis of PRs. We analyzed the transcript levels of marker genes for UPR and ERS (*Perk, Atf6, BiP, Chop*) along with markers of autophagy (*Becn1, Atg5 and Atg7*). We observed a significant decrease in the mRNA levels of *Perk* (*p* ≤ 0.05), *Atf6* (*p* ≤ 0.01) and *Chop* (*p* ≤ 0.05), whereas only the mRNA levels of *BiP* (*p* ≤ 0.05) increased significantly in AAV-treated compared to untreated retina (Figure 7A, *n* = 4). Reduced *BiP* and elevated *Chop* are typically associated with persistent stimulation of the UPR and PERK pathways in degenerative retina due to P23H rhodopsin expression [30,31]. As a key suppressor of UPR/ERS-induced apoptosis, *BiP* overexpression promotes survival pathways in PRs and reduces the rate of PR degeneration in P23H rat [31,32]. Selective activation of autophagy and increased autophagosome formation is well documented in P23H animals [33,34]. Hence, to check the status of selective autophagocytic activation and its association with autophagosome formation, we measured the levels of upstream genes (*Becn1*, *Atg5* and *Atg7*) and autophagosome-related genes (LC3B and P62/SQSTM1).

We observed a significant decrease in mRNA levels of *Becn1* (*p* ≤ 0.05), *Atg5* (*p* ≤ 0.01), and *Atg7* (*p* ≤ 0.05) as well as LC3B and P62/SQSTM1 protein expression in AAV-treated compared to untreated retinas (Figure 7A, *n* = 4). In untreated retina, a significant increase in *Becn1*, *Atg5* and *Atg7* transcript levels as well as LC3B and P62/SQSTM1 protein levels was observed in INL (Figure 7A,B). This is consistent with the selective activation of autophagy and increased autophagosome formation due to rhodopsin accumulation in the ER of rod cells in untreated retinas of P23H rats [33,34]. In contrast, we observed significantly decreased transcript levels of *Becn1*, *Atg5* and *Atg7* as well as increased protein levels of LC3B (fold change: 1.3, *p* ≤ 0.01, *n* = 4, Figure 7B(i)) and P62/SQSTM1 (fold change: 2.1, *p* ≤ 0.01, *n* = 4, Figure 7B(ii)) in AAV-treated retina. This demonstrated a significant reduction in autophagosome formation due to wild-type rhodopsin (*Rho^WT^*) trafficking from the ER to OS in the PRs of AAV-treated retinas. The expression of LC3B was restricted to OS, which indicates the formation of autophagosome vesicles to support OS recycling in AAV-treated retina. In the untreated retina, OS was largely absent but contained punctate expression of LC3B in the inner segment/outer nuclear layer (IS/ONL) and INL. Significantly increased expression of the autophagy receptor P62/SQSTM1, shown by P62-immunoreactivity indicated increased autophagy in untreated retina compared to the AAV-treated retina. Moreover, significantly increased immunoreactivity of Cathepsin D (fold change: 1.8, *p* ≤ 0.05, *n* = 4, Figure 7B(iii)) observed as green puncta was also documented in the untreated retinal regions compared to areas proximal to the AAV-injection site. Significantly increased Cathepsin-D suggests increased LMP and proteolytic degradation of autophagosomes and lysosome-mediated apoptosis of PRs.

## 4. Discussion

Retinitis pigmentosa is characterized by progressive loss of rod and cone photoreceptors, which eventually leads to blindness. Although treating adRP is challenging, recently, popularized CRISPR/Cas9-based gene editing strategies showed promising results in preclinical settings to correct adRP-associated mutations [9,10,11,12,35,36]. In this study, we utilized the AAV-CRISPR/Cas9 approach to target the most prevalent *Rho*^P23H^ mutation. We found subretinal injection to be an effective method of delivering AAV-SaCas9/gRNA into photoreceptors. In vivo *Rho*^P23H^ disruption efficiency using CRISPR/Cas9 was high in our model with no detectable off-target cleavage in genes other than *Rho*^P23H^. Selective ablation of *Rho*^P23H^ restored *Rho*^WT^ trafficking to outer segments and preserved PRs and vision by suppressing UPR/ERS-induced autophagy, which mitigated apoptotic responses. Taken together, our study demonstrated that long-term vision was preserved by the selective ablation of a dominant allele using CRISPR/Cas9.

Photoreceptor outer segments are renewed approximately every 10 days in higher vertebrates [37,38,39]. During disc morphogenesis, membrane proteins such as rhodopsin and cone opsins are synthesized in the inner segment, targeted to the outer segments, and degraded by retinal pigment epithelial (RPE) cells via phagocytosis. *Rho*^P23H^ is the most common cause of adRP in patients [40,41]. *m-**Rho*^P23H^ has been shown to cause rhodopsin misfolding and mistrafficking in rod photoreceptors, leading to ER stress and cell death [42,43]. Here, we showed that, in untreated retinas, misfolding and mistrafficking of P23H rhodopsin results in ER retention, which triggers ERS/UPR-induced autophagy responses and mediates rod photoreceptor degeneration via apoptosis. Meanwhile, *Rho*^P23H^ ablation restored proper folding and trafficking of the *Rho*^WT^ and inhibited UPR/ERS-induced autophagy-mediated apoptotic responses in rods, which provided long term protection of the retina and visual functions.

Although we documented the preservation of PRs and vision for 15 months, visual function as measured by ERG never reached to wild-type level in terms of both a- and b- wave amplitudes. Our previous study has shown that rod function was compromised at the early stage of retinal degeneration [25]. Although, in this study, gene editing targeted to rods did not rescue rod function to a normal level, progressive retinal degeneration was significantly slowed. Continued vision loss and PR degeneration were reported, not unlike that observed in current gene replacement therapies [44,45,46]. Contributing factors for efficacy deterioration overtime may involve the limitation from a single subretinal injection, which affects about one-third of the retinal area. The toxic degenerating retinal micro-environment may expose treated retina to apoptotic and necrotic signals from degenerative regions. The long-term expression of Cas9 and/or mCherry may also activate the immune system to eliminate the edited cells [47]. Although we did not detect T-cells (CD3) or macrophages (CD68) near the injection site (data not shown), we cannot exclude the possibility of exogenous proteins posing challenges to the retina that result in cell loss over time. Short-term expression of Cas9 is desired, and methods for Cas enzyme inactivation are being explored [48,49,50].

We expected low or no mCherry expression in vivo using AAV as the inverted terminal repeats (ITR) of the vector flanked the Cas9/gRNA targeting construct. However, we observed mCherry expression up to 15 months. We posited three possibilities for the observed mCherry and tested our AAV construct accordingly. First, CMV-mCherry was packaged in addition to mOp500.SaCas9-U6.gRNA.AAV since AAVs have been shown to accommodate transcripts significantly larger than 4.7 kb with reduced packaging efficiency. Second, CMV-mCherry was packaged at the expense of mOp500.SaCas9, or with lower probability by size ratio, and lastly, CMV-mCherry was packaged at the expense of U6.gRNA. Because AAV treatment disrupted *Rho*^P23H^ at the predicted site in vivo, we hypothesized that these ‘mis-packing’ scenarios occurred to a low degree from the inherent inefficiency of AAV packaging. We determined the relative abundance of vector components by real-time PCR and determined that mOp500-SaCas9 was 7 × 10^4^-fold more abundant than CMV-mCherry (Appendix A). This explains the substantively greater proportion and distribution of SaCas9 in photoreceptors compared with those that expressed mCherry. Thus, mCherry expression likely underestimated the total number of PRs in which SaCas9/gRNA was active.

The two most relevant questions for any new therapeutic approach involve the duration of the therapeutic effect and how late in the disease process the therapy can be effective. We observed retinal protection and vision rescue for at least 15 months in rat, which represents approximately 50 years in humans [51]. Our histological and functional data showed that *Rho*^P23H^-specific ablation significantly slowed down deterioration of retinal morphology and visual function. However, potential off-target cleavage events using CRISPR/Cas9 in vivo may limit the use of dominant allele ablation in translational medicine [52,53]. In a recent in vivo study, the off-target effects were detectable at low levels compared with targeted editing in clinically relevant P23H animals [11,35]. Limitations of our study included use of a transgenic model, which carries two wild-type *Rho* alleles in addition to *Rho*^P23H^. Although this does not accurately represent the situation in patients who would presumptively be left with one functional *RHO* allele, the problem for haploinsufficiency has not been reported. Individuals bearing one functional *RHO* allele are frequently asymptomatic. Furthermore, *RHO*^P23H^ hemizygosity with 13% mosaicism was observed in a middle-aged individual with normal vision, suggesting that P23H ablation in a moderate-fraction PR is sufficient to prevent significant loss of PRs and functional vision. With current gene editing technology, animal model testing is still meaningful, as it provides a window to study the safety and efficacy of CRISPR/Cas9 as well as other in vivo gene editing techniques.

Increasing gene editing efficacy and AAV distribution by multiple injections may further improve therapeutic efficiency by reducing the negative impact from untreated degenerative retina. Other delivery modalities, such as intravitreal injections, may provide greater distribution. Delivery of Cas9 mRNA permits transient expression, which limits the potential for off-target effects and long-term expression of Cas9. Furthermore, determining the underlying multi-omics landscape of edited retinal cells may identify key transcriptional changes that promote long-term PR survival and the spatio-temporal transcriptomic changes that occur in surrounding retinal cells, such as secondary horizonal cells, bipolar cells, projecting ganglion cells, and RPE. Advancing the knowledge of the host environment may help to optimize treatments for inherited retinal diseases.

## Figures and Tables

**Figure 1 pharmaceutics-14-00824-f001:**
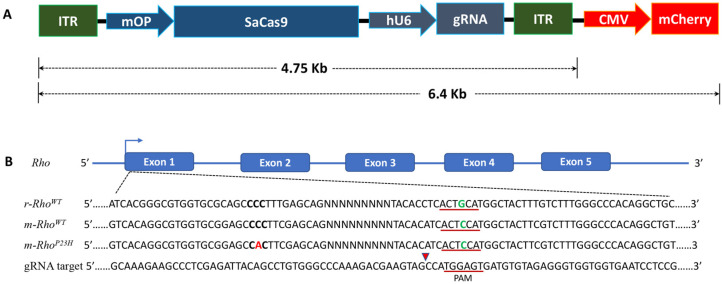
CRISPR/Cas9-mediated m-*Rho*^P23H^ ablation in vivo. (**A**) Illustration of CRISPR vector; pAAV2-mOp-SaCas9-U6-gRNA-CMV-mCherry. (**B**) Rat and mouse wild-type (WT) *Rho* with exon 1 showing the P23H point mutation (**C**→**A**) in m-*Rho* (bold and red font). The gRNA targeting and PAM sequences are underlined. Mismatches in the PAM sequences are shown in green font. Red arrowhead indicates the predicted cleavage site in *m-**Rho^P23H^*.

**Figure 2 pharmaceutics-14-00824-f002:**
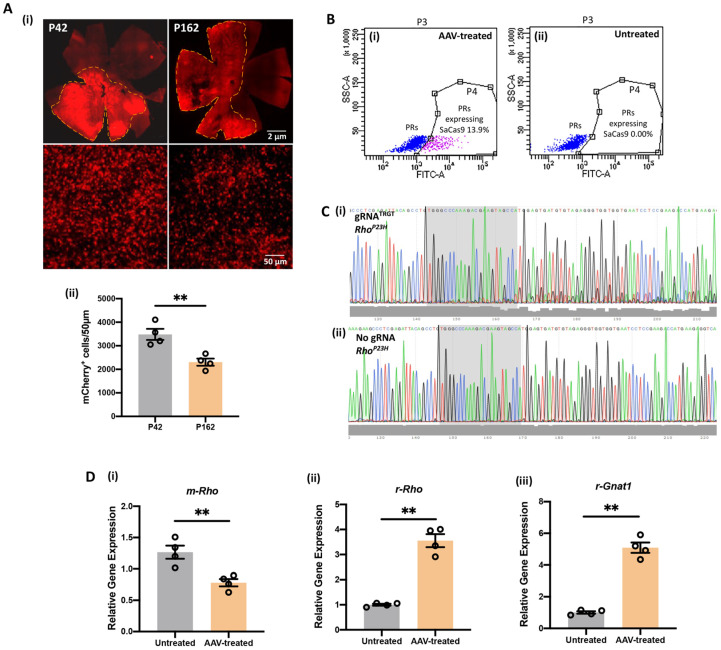
Assessment of CRISPR/Cas9-edited photoreceptors in vivo. (**A**) (**i**)**.** Representative retinal whole mount images (top panel) show broad distribution of mCherry expression at P42 and P162. The retina samples were taken from P23H rats that received a single unilateral trans-scleral subretinal injection of 2 µL AAV-SaCas9/gRNA at P24. (**ii**) Bottom panel shows representative images of mCherry expression in photoreceptors. Scale bar, 2 mm for the top panel whole mount images, and 50 µm for the bottom panel images. (**B**) High side-scatter of heterochromatin-identified photoreceptor perikarya (blue) in AAV-injected (**i**) and untreated eyes (**ii**) from each animal. PRs that expressed SaCas9 were sorted from AAV-injected eyes by SaCas9-HA fusion protein expression using HA.11-FITC (**i**): population shown in pink: 13.9%, which was absent in contralateral uninjected (control) eyes (**ii**): population shown in pink: 0.0%. (**C**) Following genomic DNA extraction and PCR amplification of the predicted cleavage site, Sanger sequencing confirmed *Rho*^P23H^ disruption downstream of the gRNA binding site in treated eyes (**i**), shaded region, but not in uninjected (control) eyes (**ii**). The presence of overlapping base calls with decreased Phred quality scores (grey bars) indicates genomic variants generated by indels during DNA repair following NHEJ. (**D**) (**i**–**iii**) qPCR of mouse-*Rho* (*m-Rho*), rat-*Rho* (*r-Rho*) and rat-*Gnat1* (*r-Gnat1*) in AAV-treated and untreated retina. Data represent the mean ± SEM. ** *p ≤* 0.01.

**Figure 3 pharmaceutics-14-00824-f003:**
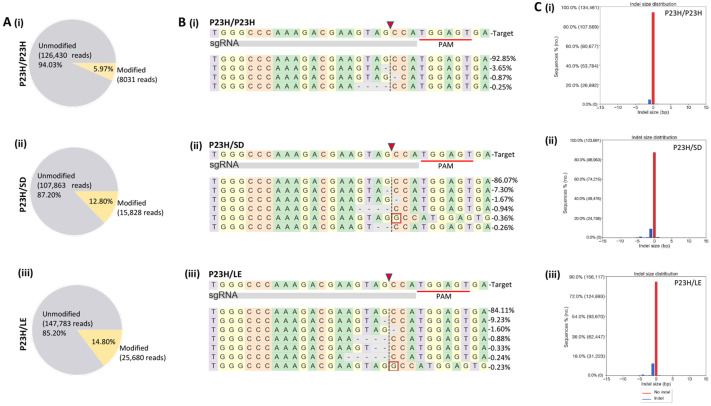
On-target analysis of CRISPR/Cas9-editing in AAV-treated P23H rats. (**A**) Representative Pie charts show the editing efficiency of 5.97% in P23H/P23H (**i**), 12.08% in P23H/SD (**ii**) and 14.8% in P23H/LE (**iii**) rats. (**B**) (**i**–**iii**) Deep sequencing reads (presented as percent score) show insertions/deletions (indels) at the predicted cleavage site (red arrowhead). The sequence with high reads score is the unmodified sequence, and underlined are the PAMs. The insertions are indicated in the red box and deletions with a dash (-). (**C**) Representative graphs show indel size distribution in P23H/P23H (**i**), P23H/SD (**ii**) and P23H/LE (**iii**) rats.

**Figure 4 pharmaceutics-14-00824-f004:**
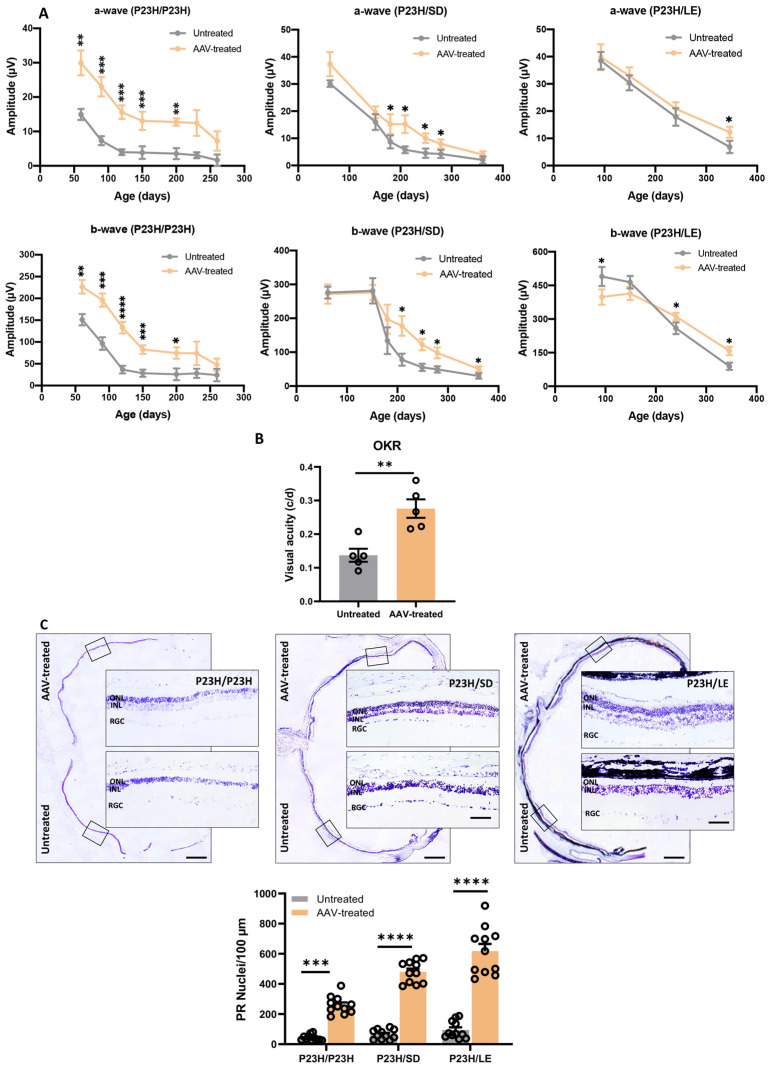
Visual function assessment and histological evaluation of AAV-treated P23H rats. (**A**) Functional evaluation of retinas from P23H/P23H, P23H/SD and P23H/LE rats. ERG was used to evaluate the retinal function of P23H/P23H (*n* = 12) from P60-P260, P23H/SD (*n* = 8) from P60-P360 and P23H/LE (*n* = 12) from P90-P350. Progressive decrease in a- and b-waves was observed in AAV-treated and untreated eyes. However, AAV-treated eyes exhibited significantly higher ERG responses over a longer period, suggesting that the rescued photoreceptors contributed to visual function. (**B**) Optokinetic response (OKR) was tested in P23H/LE rats (*n* = 5) at 15 months of age. AAV-treated eyes scored significantly higher than untreated fellow eyes. At P450, visual acuity was maintained at over 50% of the value of wild-type rats. (**C**) Retinal montage images of cresyl violet-stained retina showing histological structure at the AAV-injection site and contralateral untreated site. The retina samples were collected from P23H/P23H, P23H/SD and P23H/LE rats, respectively at P450. Significant long-term preservation of photoreceptors is evident in AAV-treated areas compared to contralateral regions distal to the injection site that were devoid of mCherry expression. Few or no photoreceptors remained in these regions due to advanced retinal degeneration. High power images show preserved photoreceptors vs. degenerative retina distal to the injection site. At P450, significant survival of photoreceptors was observed in the ONL of P23H/P23H (1–3 layers), P23H/SD (1–4 layers) and P23H/LE (4–5 layers) rats that received AAV injection (wild-type rat has 10 layers of ONL). Scale bar, 800 µm for the retina montage image and 100 µm for the amplified image. INL, inner nuclear layer; ONL, outer nuclear layer; PR, photoreceptor; RGC, retinal ganglionic cell. Data represent the mean ± SEM. * *p ≤* 0.05, ** *p ≤* 0.01, *** *p ≤* 0.001, **** *p ≤* 0.0001.

**Figure 5 pharmaceutics-14-00824-f005:**
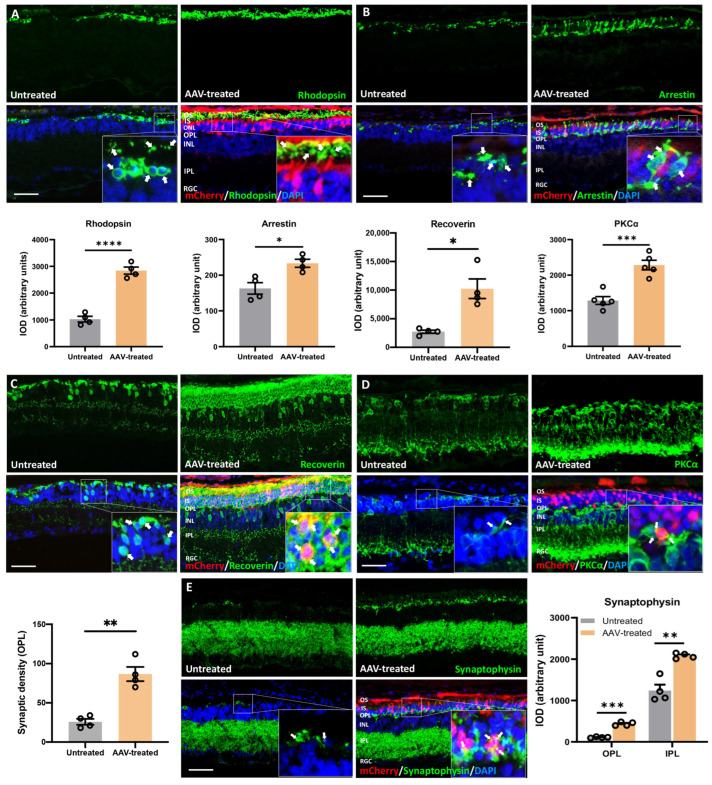
Preservation of rod and cone photoreceptors, rod-bipolar dendritic arborization, and retinal synapses following m-*Rho*^P23H^ ablation. Retinal sections from P450 P23H rats were processed for immunofluorescence (IF) using antibodies against rhodopsin (**A**), cone arrestin (**B**), recoverin (**C**), PKCα (**D**) and synaptophysin (**E**). (**A**) Rhodopsin IF image shows significantly increased expression of (green) PR outer segments (OS) (yellow: green; colocalized with mCherry (red): yellow) in retinal regions proximal to AAV injection compared to the contralateral retina distal to the injection site. In these regions, rhodopsin was restricted to cell bodies, which suggests ER retention of rhodopsin. (**B**) Arrestin IF image of AAV-treated retina shows significantly increased immunofluorescence intensity (green) and number of immunolabeled cones with non-pathological morphological features. Cones were infrequently observed in untreated retinas and were fragmented and irregular shaped, consistent with apoptotic bodies, suggesting that CRISPR ablation therapy rescued cone PRs. (**C**) Recoverin IF image shows significantly increased immunoreactivity (green) in PRs and bipolar cells in AAV-treated retina, suggesting significant PR preservation in the ONL along with rescue and non-pathological morphology of bipolar cells. (**D**) PKCα IF image shows rescue of rod bipolar cells and significantly elaborated rod-bipolar dendritic arborization in the INL and OPL layers at AAV-treated sites. Further, synaptic connections between bipolar cells and mCherry expressing rod cells was observed in AAV-treated retina, in contrast with fewer synaptic connections in untreated retinas. (**E**) Synaptophysin IF image shows significantly increased staining (**green**) and number of synaptic puncta in OPL and IPL in AAV-treated areas compared to the untreated areas, suggesting that gene editing preserved retinal synapses. Scale bar, 100 µm. Nuclei were stained with DAPI. INL, inner nuclear layer; IS, inner segment; ONL, outer nuclear layer; OS, outer segment; PR, photoreceptor; RGC, retinal ganglionic cell. Data represent the mean ± SEM (*n* = 4–5). * *p ≤* 0.05, ** *p ≤* 0.01, *** *p ≤* 0.001, **** *p ≤* 0.0001.

**Figure 6 pharmaceutics-14-00824-f006:**
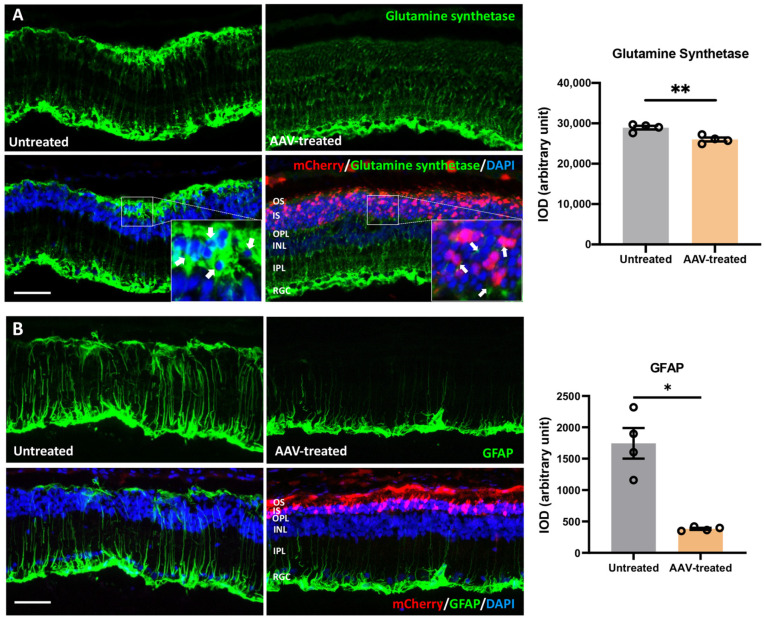
m-*Rho*^P23H^ ablation preserved retinal morphology and prevented Müller glia activation. (**A**) The areas within the retinas distal to the injection site had thick and elongated processes of Müller glia (green: GS^+^) that spanned the ONL and surrounded condensed nuclei (DAPI), consistent with degenerating photoreceptors (arrows). AAV-treated retina shows normal morphology of Müller glia with single and thin processes that only span the ONL. (**B**) Retinal regions distal to AAV injection show significantly increased GFAP immunoreactivity (green) indicative of activated Müller glia, a hallmark of Müller cell gliosis. The number of active Müller glia was reduced significantly from AAV treatment, indicated by decreased GFAP expression that was restricted to end feet. Scale bar, 100 µm. Nuclei were stained with DAPI. INL, inner nuclear layer; IS, inner segment; ONL, outer nuclear layer; OS, outer segment; PR, photoreceptor; RGC, retinal ganglionic cell. Data represent the mean ± SEM (*n* = 4). * *p ≤* 0.05, ** *p ≤* 0.01.

**Figure 7 pharmaceutics-14-00824-f007:**
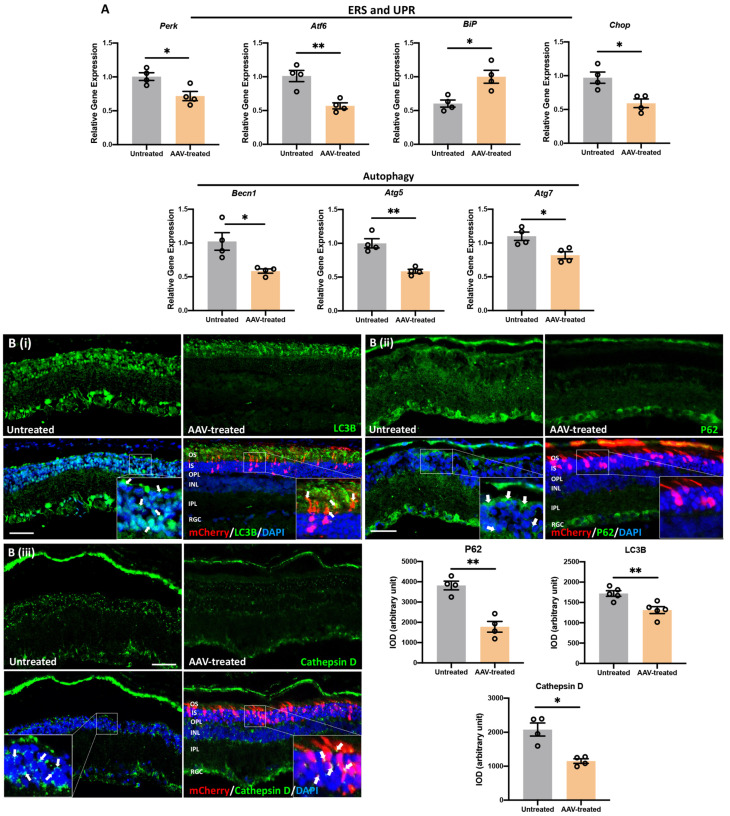
m-*Rho*^P23H^ ablation reduced retinal autophagy via suppressing UPR and ERS. (**A**) qPCR of *Perk, BiP, Atf6, Chop* (marker genes of UPR and ERS) and *Becn1, Atg5, Atg7* (marker genes of autophagy). In AAV-treated retinas, only the BiP functional transcripts showed significantly increased expression, whereas mRNA levels of remaining genes associated with UPR/ERS and autophagy decreased significantly compared to untreated retina. (**B**) Immunofluorescence of (**i**) autophagosomes (LCB3), (**ii**) autophagy receptors (P62/SQSTM1) and (**iii**) lysosomes (cathepsin D) in AAV-treated vs. untreated retinas. LC3B, P62 and cathepsin-D immunoreactivity was decreased significantly in AAV-treated retina compared to the untreated retina. LC3B expression was restricted to the outer segments of the PR layer in AAV-treated retina. By contrast, untreated retina showed punctate expression of LC3B, P62, and cathepsin D in the INL. Significantly reduced expression of P62 and cathepsin D was observed in RPE, ONL, and INL of AAV-treated retina. Scale bar, 100 µm. Nuclei were stained with DAPI. INL, inner nuclear layer; IS, inner segment; ONL, outer nuclear layer; OS, outer segment; PR, photoreceptor; RGC, retinal ganglionic cell. Data represent the mean ± SEM (*n* = 4). * *p ≤* 0.05, ** *p ≤* 0.01.

## Data Availability

All the data reported in this study are shown in this manuscript and in Appendix A.

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
