# Peer review of "AAV-CRISPR/Cas9 Gene Editing Preserves Long-Term Vision in the P23H Rat Model of Autosomal Dominant Retinitis Pigmentosa"

_pharmaceutics, 2022, doi:10.3390/pharmaceutics14040824_

Round 1

Reviewer 1 Report

Retinitis Pigmentosa (RP) is one of the most common inherited retinal diseases caused by variants in more than 60 genes. Major complain of RP is the progressive loss of vision. As discussed by the authors, autosomal recessive forms of RP benefits from FDA approved gene therapy but there are limited options for autosomal dominant forms that require reduction of deleterious protein levels. The authors have addressed this using a slow degeneration adRP rat model with rhodopsin P23H mutation. In this model photoreceptor degeneration starts from P15 permitting the authors to test treatments at multiple timeframes. The authors use a smaller SaCas9 that can be packaged into AAV2/8 and the gRNA delivery can be verified using mCherry reporter. Post subretinal delivery, both functional rescue and structural preservations. I have several concerns regarding the data and interpretation of the results.

  • Sex as a biological attribute is not considered in the paper.
  • The contralateral eye is considered as sham-operated or untreated control. The controls in the experimental control also should reflect this clearly.
  • Both for OKR and ERG wildtype animals were not used as positive controls.
  • Only one stimuli 25 cd/m^2 was tested for ERG which reflects highest flash intensity the authors should also report results at least two or three time points.
  • Similarly, in comparing ERG results, there are no data on Wildtype rats because compared to wildtype rat a-wave the magnified increase in a- and b-wave responses rescued here with the treatment might not be insignificant.
  • Experimental evidence needs to include allele specific targeting and editing by the chosen guide RNA. As an example, Takara Bio provides the guide-it genotype confirmation kit is a streamlined, in vitro alternative to tedious subcloning and sequencing following CRISPR/Cas9 editing of target genes
  • Histologic assessment should confirm validation and specificity of the used antibodies.
  • For imaging results the authors should include images from wildtype retina as this will help readers who are not knowledgeable about retina structure to appreciate the rescue of structure after genome editing.
  • How is the percentage editing measured, please detail in the methods as to the calculations is the percentage of positive cells calculated just within the retinal area that shows reporter, if so please calculate total number of cells and number of cells expressing reporter. Also, this reviewer is surprised that AAV2/8 only transduces areas around the injection site.
  • Please elaborate on the rationale for testing mouse Rho expression and that too it shows a reduction after AAV-treated.
  • ERG results need to be elaborated as only some rescue was seen in P23H/P23H retina but not in other two models.
  • Since the functional results does not demonstrate long-term vision rescue this should be down played throughout the manuscript.
  • The authors have to be careful about the discussion points. The use of CMV promoter for gene expression in photoreceptors proved that photoreceptors are able to silence CMV promoter. How does this explain the results of mCherry or even long-term expression of Cas9 and guide RNA?
  • It is surprising that the authors missed an opportunity to use a PR specific promoter to justify clinical translation.

Author Response

Retinitis Pigmentosa (RP) is one of the most common inherited retinal diseases caused by variants in more than 60 genes. Major complain of RP is the progressive loss of vision. As discussed by the authors, autosomal recessive forms of RP benefits from FDA approved gene therapy but there are limited options for autosomal dominant forms that require reduction of deleterious protein levels. The authors have addressed this using a slow degeneration adRP rat model with rhodopsin P23H mutation. In this model photoreceptor degeneration starts from P15 permitting the authors to test treatments at multiple timeframes. The authors use a smaller SaCas9 that can be packaged into AAV2/8 and the gRNA delivery can be verified using mCherry reporter. Post subretinal delivery, both functional rescue and structural preservations. I have several concerns regarding the data and interpretation of the results.

Point 1: Sex as a biological attribute is not considered in the paper.

Response 1: Thank you for your comments, we have added the animal sex information in the text. Yes, both male and female rats were used for this study.

Point 2: The contralateral eye is considered as sham-operated or untreated control. The controls in the experimental control also should reflect this clearly.

Response 2: Thank you! We did have control injections and untreated control. We have edited method section accordingly.

Point 3: Both for OKR and ERG wildtype animals were not used as positive controls.

Response 3: Good points! We have added data from wildtype rats as comparison.

Point 4: Only one stimuli 25 cd/m^2 was tested for ERG which reflects highest flash intensity the authors should also report results at least two or three time points.

Response 4: Thank you for suggestion. We did use different stimuli in our other studies. We will use more than one stimuli in future study.

Point 5: Similarly, in comparing ERG results, there are no data on Wildtype rats because compared to wildtype rat a-wave the magnified increase in a- and b-wave responses rescued here with the treatment might not be insignificant.

Response 5: Your valuable point is well taken. The main purpose of this study is to test vision preservation after ablating mouse Rho. From this long-term study, we have seen photoreceptor and visual function preservation significantly better than controls, but not reach the same level as the wild type. Since the treatment started when there is ongoing retinal degeneration; a single subretinal injection only affect about 1/3 of the retinal area, leaving the rest of retina undergoes progressive retinal degeneration. We have discussed this in the discussion.

Point 6: Experimental evidence needs to include allele specific targeting and editing by the chosen guide RNA. As an example, Takara Bio provides the guide-it genotype confirmation kit is a streamlined, in vitro alternative to tedious subcloning and sequencing following CRISPR/Cas9 editing of target genes;

Response 6: Thank you for reviewer’s advice.

Point 7: Histologic assessment should confirm validation and specificity of the used antibodies.

Response 7: Thank you for your suggestion, we did perform antibody validation and specificity, these have been added to the method section.

Point 8: For imaging results the authors should include images from wildtype retina as this will help readers who are not knowledgeable about retina structure to appreciate the rescue of structure after genome editing.

Response 8: Thank you for reviewer’s suggestion. We have added ONL thickness in the text.

Point 9: How is the percentage editing measured, please detail in the methods as to the calculations is the percentage of positive cells calculated just within the retinal area that shows reporter, if so please calculate total number of cells and number of cells expressing reporter. Also, this reviewer is surprised that AAV2/8 only transduces areas around the injection site.

Response 9: We have incorporated the details in the Materials and Methods section under the heading Retinal cell isolation, FACS-based sorting and Targeted deep sequencing. Please see the response of 3rd comment for AAV2/8 transduce area.

Point 10: Please elaborate on the rationale for testing mouse Rho expression and that too it shows a reduction after AAV-treated.

Response 10: We have elaborated the rational for testing mouse Rho expression in text. Since Rho ablation by CRISPR/Cas9 only affects part of the retina. We wanted to test mouse Rho expression after ablation. Yes, our result shows mouse Rhoexpression was reduced after treatment.

Point 11: ERG results need to be elaborated as only some rescue was seen in P23H/P23H retina but not in other two models.

Response 11: Thanks reviewers’ comment. We have edited the text.

Point 12: Since the functional results does not demonstrate long-term vision rescue this should be down played throughout the manuscript.

Response 12: Thanks reviewers’ comment. We have shown there is long-term photoreceptor and vision preservation after Rho ablation, although there is need for improvement in the future study as we stated in discussion.

Point 13: The authors have to be careful about the discussion points. The use of CMV promoter for gene expression in photoreceptors proved that photoreceptors are able to silence CMV promoter. How does this explain the results of mCherry or even long-term expression of Cas9 and guide RNA?

Response 13: Thanks for reviewers’ comments. SaCas9 was driven by photoreceptor specific promoter-mouse rhodopsin, mCherry was driven by CMV promoter.

Point 14: It is surprising that the authors missed an opportunity to use a PR specific promoter to justify clinical translation.

Response 14: See our answer above.

Reviewer 2 Report

AAV gene therapy has been utilized to treat monogenic disorders such as Leber’s congenital amaurosis (LCA), which is a group of eye diseases, and in RPE65-LCA the mutations in the RPE65 gene results in retinal dystrophy. Therapeutic efforts target delivery of the corrected gene via an AAV vector to replace the mutant protein and restore vision. This strategy has been successfully applied to LCA and Luxturna received US FDA approval for its use in the clinic for Retinitis Pigmentosa (RP). However, other types of RP result from autosomal dominant mutations, so these therapeutic efforts target reducing the amount of protein to reduce the toxic effects of these mutant proteins. In this paper, the authors employ CRISPR/Cas9 gene editing technology delivered by an AAV2/8 vector by subretinal injection in the P23H rat model of autosomal dominant RP. They evaluated safety and demonstrated that subretinal delivery rescued photoreceptors and vision for up to 15 months in these animals. CRISPR technology for gene editing may be a useful strategy for this disorder as well as other autosomal dominant disorders. After m-RhoP23H ablation, key signaling molecules in the rod phototransduction cascade were increased suggesting restoration of the signaling cascade. They also evaluated off-target effects, which is one of the concerns in the use of CRISPR/Cas9 technology. In this study, in the regions evaluated, no off-target effects occurred. The AAV-CRISPR/Cas9 approach showed promising results in treating this autosomal dominant condition and may also be useful for other autosomal dominant disorders. This approach in vivo also may allow for studies of potential immune responses to treatment prior to moving forward into clinical trials.

Revisions needed:

Page 2, Line 55 – comma instead of period after (age 3 – 7)

Page 9, Line 324/325 – change the word functional to function or functionality.

Author Response

AAV gene therapy has been utilized to treat monogenic disorders such as Leber’s congenital amaurosis (LCA), which is a group of eye diseases, and in RPE65-LCA the mutations in the RPE65 gene results in retinal dystrophy. Therapeutic efforts target delivery of the corrected gene via an AAV vector to replace the mutant protein and restore vision. This strategy has been successfully applied to LCA and Luxturna received US FDA approval for its use in the clinic for Retinitis Pigmentosa (RP). However, other types of RP result from autosomal dominant mutations, so these therapeutic efforts target reducing the amount of protein to reduce the toxic effects of these mutant proteins. In this paper, the authors employ CRISPR/Cas9 gene editing technology delivered by an AAV2/8 vector by subretinal injection in the P23H rat model of autosomal dominant RP. They evaluated safety and demonstrated that subretinal delivery rescued photoreceptors and vision for up to 15 months in these animals. CRISPR technology for gene editing may be a useful strategy for this disorder as well as other autosomal dominant disorders. After m-RhoP23H ablation, key signaling molecules in the rod phototransduction cascade were increased suggesting restoration of the signaling cascade. They also evaluated off-target effects, which is one of the concerns in the use of CRISPR/Cas9 technology. In this study, in the regions evaluated, no off-target effects occurred. The AAV-CRISPR/Cas9 approach showed promising results in treating this autosomal dominant condition and may also be useful for other autosomal dominant disorders. This approach in vivo also may allow for studies of potential immune responses to treatment prior to moving forward into clinical trials.

Response: Thank you very much for reviewer’s comments.

Revisions needed:

Page 2, Line 55 – comma instead of period after (age 3 – 7)

Page 9, Line 324/325 – change the word functional to function or functionality.

Response: Thank you! Changes have been made accordingly.

Round 2

Reviewer 1 Report

Thank you for addressing some of the review comments but the authors have responded with only cursory changes to the reviews without any attempt to support with additional data. This is an unfortunate response by the authors. 

As an example, the standardized ERG protocol for studying retinal function has been discussed several times and through several previous publications as the retinal function in rodents depends on the design of the electrode with the stimulus protocol a secondary consideration. The authors pay no attention to this, and even after suggested by the reviewer. 

Author Response

Response:

Thank you for your valuable comments. Yes, the ERG protocol for studying retinal function has been discussed in previous studies. In fact, our previous study has characterized this P23H line 1 rat model for retinitis pigmentosa. Our study showed rod function was compromised at early stage of retinal degeneration (postnatal day 28)1. We have added new ERG data from wild type rats as comparison with the same light stimulation protocol as we used in this manuscript. The data has been added to supplement figure S3.  In the wild type rats, a-wave is 125.96 ±11.26 µV and b-wave is 582.26 ±14.69 µV. In our current study, gene-editing by CRISPR/Cas9 did not preserve rod function to wild type level (Figure 4, P23H/LE), but  slowed down progressive cone degeneration significantly. The reason for not being able to rescue rod function at normal level after specific gene editing on rods is likely due to the followings: 1). It is well established that a single subretinal injection of AAV-vector only affects about 1/3 of retina area, leaving majority of the retina undergoes progressive degeneration. 2). Since the cutting efficacy by CRISPR/Cas9 was about 5.9-14.8% (Figure 3) from mCherry+ photoreceptors, the response from these edited rod photoreceptors probably too low to be recorded by ERG. However, this study showed significantly better ERG a-and b-wave amplitudes were achieved over one year after CRISPR/Cas9 gene editing.

We have edited text with new ERG data from wild type rats as comparison.

 1.     Bin Lu et al. Retinal morphological and functional changes in an animal model of retinitis pigmentosa. Vis Neurosci. 2013 May; 30(3):77-89.

Round 3

Reviewer 1 Report

Agreed with the supporting data